# Voluntary Wheel Running Mitigates Disease in an Orai1 Gain-of-Function Mouse Model of Tubular Aggregate Myopathy

**DOI:** 10.3390/cells14171383

**Published:** 2025-09-04

**Authors:** Thomas N. O’Connor, Nan Zhao, Haley M. Orciuoli, Sundeep Malik, Alice Brasile, Laura Pietrangelo, Miao He, Linda Groom, Jennifer Leigh, Zahra Mahamed, Chen Liang, Feliciano Protasi, Robert T. Dirksen

**Affiliations:** 1Genetics and Genomics Graduate Program, Department of Biomedical Genetics, University of Rochester Medical Center, Rochester, NY 14642, USA; 2Department of Pharmacology and Physiology, University of Rochester Medical Center, Rochester, NY 14642, USA; 3Department of Biology, Biological Sciences, University of Rochester, Rochester, NY 14642, USA; 4Center for Advanced Studies and Technology (CAST), University G. d’Annunzio of Chieti-Pescara, I-66100 Chieti, Italy; 5Department of Medicine and Aging Sciences (DMSI), University G. d’Annunzio of Chieti-Pescara, I-66100 Chieti, Italy

**Keywords:** myopathy, exercise, proteomics, skeletal muscle, aggregate, mitochondria, physiology, sarcoplasmic reticulum

## Abstract

Tubular aggregate myopathy (TAM) is an inherited skeletal muscle disease associated with progressive muscle weakness, cramps, and myalgia. Tubular aggregates (TAs) are regular arrays of highly ordered and densely packed straight-tubules observed in muscle biopsies; the extensive presence of TAs represent a key histopathological hallmark of this disease in TAM patients. TAM is caused by gain-of-function mutations in proteins that coordinate store-operated Ca^2+^ entry (SOCE): STIM1 Ca^2+^ sensor proteins in the sarcoplasmic reticulum (SR) and Ca^2+^-permeable ORAI1 channels in the surface membrane. Here, we assessed the therapeutic potential of endurance exercise in the form of voluntary wheel running (VWR) in mitigating TAs and muscle weakness in *Orai1^G100S/+^* (GS) mice harboring a gain-of-function mutation in the ORAI1 pore. Six months of VWR exercise significantly increased specific force production, upregulated biosynthetic and protein translation pathways, and normalized both mitochondrial protein expression and morphology in the *soleus* of GS mice. VWR also restored Ca^2+^ store content, reduced the incidence of TAs, and normalized pathways involving the formation of supramolecular complexes in fast twitch muscles of GS mice. In summary, sustained voluntary endurance exercise improved multiple skeletal muscle phenotypes observed in the GS mouse model of TAM.

## 1. Introduction

Calcium (Ca^2+^) is a universal second messenger that regulates a multitude of cellular processes in addition to its integral role in linking skeletal muscle excitation to force production. As a result, even minor disruptions in the proper control of Ca^2+^ storage, release, and reuptake can lead to profound effects on cellular stress levels, gene expression, and muscle performance. The sarcoplasmic reticulum (SR) serves as the main intracellular site for Ca^2+^ storage and release in skeletal muscle, dynamically regulating cytosolic Ca^2+^ concentrations that coordinate actin-myosin cross-bridge cycling and subsequently muscle force production [1]. Upon excitation of skeletal muscle, stored Ca^2+^ is released from the SR via depolarization-induced conformational changes in the dihydroypyridine receptor (DHPR) voltage sensor located in the transverse tubule membrane that is mechanically coupled to Ca^2+^ release channels, or ryanodine receptors (RyR1), in the terminal cisternae of the SR [2,3]. While Ca^2+^ entry through the DHPR does not significantly contribute to elevation in myoplasmic Ca^2+^ during EC coupling [4,5,6], SR Ca^2+^ store depletion during repetitive high-frequency stimulation activates Ca^2+^ entry via store-operated Ca^2+^ entry (SOCE) channels. As in non-excitable cells, SOCE in skeletal muscle is also coordinated by stromal interaction molecule 1 (STIM1) luminal SR Ca^2+^ sensor proteins and Ca^2+^ selective ORAI1 Ca^2+^ channels located in the transverse tubule membrane [7,8,9]. Proper SOCE, and subsequently proper STIM1 and ORAI1 function, play key roles in muscle development [8,10,11], fatigue resistance [9,12,13,14], and maintaining muscle function with age [15], in addition to playing key roles in various systemic processes [16,17,18]. N-terminal EF-hand domains of STIM1 typically bind luminal SR Ca^2+^ when stores are replete; store depletion causes conformational changes to STIM1, enabling oligomerization, relocalization to junctional SR closer in proximity to the plasma membrane and activation of the Ca^2+^-selective ORAI1 channel [19,20,21]. Mutations in STIM1 and ORAI1 are linked to a multitude of muscle-related disorders with varying levels of severity, age of onset, and multi-systemic involvement [17,22,23]. Recessive loss-of-function mutations that lead to a reduction in SOCE activity result in severe combined immunodeficiency, autoimmunity, ectodermal dysplasia, mydriasis, hypotonia and muscle weakness [24,25]. Conversely, dominant gain-of-function mutations in the SOCE machinery cause tubular aggregate myopathy (TAM) and Stormorken Syndrome, a clinical continuum characterized by myalgia, muscle cramps, and progressive muscle weakness in addition to thrombocytopenia, hyposplenism, ichthyosis, miosis, short stature, and dyslexia [18,26,27,28,29,30,31,32,33,34,35]. TAM-causing mutations lead to constitutive ORAI1 channel activation and excessive Ca^2+^ entry causing an increase in cytosolic Ca^2+^ concentration [26,27,31,33]. Currently, there is no cure or effective treatment for TAM, although preclinical pharmacologic [36,37] and genetic [38] inhibition of ORAI1 has shown limited promise.

A major hallmark of TAM is the presence of highly ordered and densely packed SR straight-tubes aligned in honeycomb-like structures, referred to as tubular aggregates (TAs). Importantly, TAs typically display fiber type preference, forming primarily in fast twitch, glycolytic type IIb/IIx fibers [26,39,40,41,42], and are more frequently observed in males [40,42]. Interestingly, TAs are also routinely observed in fast twitch skeletal muscle of aged (e.g., two-year-old) male, but not female, mice [39,40]. TAs contain large amounts of Ca^2+^ [41] and are of SR origin as they stain positively for STIM1 and other SR proteins including sarco/endoplasmic reticulum Ca^2+^-ATPase (SERCA), calsequestrin 1 (CASQ1), RYR1, triadin and sarcalumenin [26,29,40,43]. Recent findings identified novel variants in the CASQ1 and RYR1 genes in patients with TAM [44,45,46,47] suggesting that TAs may represent compensation designed to sequester Ca^2+^ resulting from defects in SR Ca^2+^ handling. Several recently generated gain-of-function STIM1 TAM mouse models (D84G, I115F, R304W) recapitulate key multi-systemic features of the disease but lack the key histopathological formation of TAs in skeletal muscle [18,48,49]. In contrast, we recently found that *Orai1^G100S/+^* (GS) knock-in mice exhibit an age-dependent myopathy characterized by muscle weakness, elevated serum creatine kinase and robust presence of TAs in fast twitch muscles (*extensor digitorum longus*, EDL; *flexor digitorum brevis*, FDB; and *tibialis anterior*), which recapitulate key clinically relevant observations observed in TAM patients with analogous mutation in human ORAI1 (G98S) [26,43,50]. Additionally, Pérez-Guàrdia et al., recently developed *Orai1^V109M/+^* (VM) knock-in mice that recapitulate several myopathic (muscle weakness, TA presence in the *tibialis anterior*) and systemic (smaller size, spleen enlargement, thrombocytopenia) effects observed in TAM patients with analogous mutation in human ORAI1 (V107M) [51]. Thus, GS and VM knock-in mice represent the first TAM mouse models that exhibit TAs [52].

While multifaceted, endurance exercise drives changes in skeletal muscle that ultimately lead to improved muscle function. Boncompagni et al. demonstrated that sustained voluntary endurance exercise in aging mice (voluntary wheel running, VWR) both restores SOCE function and prevents the formation of TAs that are typically observed in fast twitch EDL muscle of two-year-old male mice [53]. Consistent with these findings, sustained voluntary endurance exercise results in reduced expression of CASQ1, SERCA, and mitochondrial calcium uniporter (MCU), and increased expression of sarcolemmal Ca^2+^ export mechanisms (Na^+^/Ca^2+^ exchanger, NCX and plasma membrane Ca^2+^ ATPase, PMCA) [54]. A decrease in expression of CASQ1 and SERCA with exercise is consistent with the observed exercise-induced reduction in TAs observed in muscle of aged male mice. Decreased MCU expression with exercise would be expected to protect against mitochondrial Ca^2+^ overload and damage, in conjunction with increased extrusion of myoplasmic Ca^2+^ as a result of increased expression of PMCA and NCX. Thus, we hypothesize that sustained voluntary endurance exercise would provide similar beneficial adaptive changes in sarcolemmal, SR, and mitochondrial Ca^2+^ influx/efflux balance in muscle of GS TAM mice that protect against Ca^2+^ overload and formation of TAs that preserves muscle function. Additionally, we posit that exercise-induced increases in protein translation and turnover will aid in the degradation and replacement of damaged proteins and organelles that can occur in myopathic muscle [55,56].

The objective of this study was to characterize the impact of sustained voluntary endurance exercise on the TAM phenotype of GS knock-in mice. In sedentary GS mice, we observed impaired contractile function (both in EDL and *soleus*) that coincides with disrupted Ca^2+^ dynamics, TAs in fast twitch muscle (EDL), and ultrastructural changes in sarcomere/mitochondria (*soleus*). These changes paralleled distinct alterations in the EDL and *soleus* muscle proteomes. Six months of voluntary endurance exercise (VWR) significantly improved GS mouse *soleus* contractile function, reduced TAs in EDL muscles, and normalized the major changes observed in the EDL and *soleus* muscle proteomes.

## 2. Materials and Methods

### 2.1. Animals

This study was carried out in accordance with the recommendations in the Guide for the Care and Use of Laboratory Animals of the National Institutes of Health. All procedures involving animals were approved by the Institutional Animal Care and Use Committee (IACUC) at the University of Rochester called the University Committee on Animal Resources (UCAR). *Orai^G100S/+^* and *Orai1^V5HA/+^* mice were generated by the University of Rochester Transgenic and Gene Editing Core Facility. To assess ORAI1 expression in skeletal muscle, *Orai1^V5HA/+^* mice were crossed with *Orai1^G100S/+^* mice to generate compound heterozygous *Orai1^G100S/V5HA^* mice. Adult C57BL/6J (two to eight months old, The Jackson Laboratory, Bar Harbor, ME, USA) mice and were singly housed with low-profile running wheels (see below for details) at eight weeks of age in accordance with an approved UCAR protocol. All data reported in this study were from terminal experiments conducted on eight-month-old male and female mice unless otherwise stated. All mice were maintained on a 12:12 light/dark cycle and provided ad libitum access to pelleted feed and standard drinking water (Hydropac, Avidity Science, Waterford, WI, USA).

### 2.2. Voluntary Wheel Running Exercise

Low profile wireless rodent wheels (ENV-047 wheels, Med Associates, Fairfax, VT, USA) were used in singly housed mouse cages to track voluntary running activity over a 6-month period (from two to eight months of age), with sedentary control animals being singly housed in cages with locked wheels that were unable to rotate. Wheels were connected to a wireless central hub that recorded total running activity using Wheel Manager software (SOF-860, Med Associates, Fairfax, VT, USA) and data analyzed using Wheel Analysis software (SOF-861, Med Associates, Fairfax, VT, USA) reported as total distance run per day (km/day). Terminal experiments were performed immediately after removing mice from cages with wheels. In order to reduce experimental variability, only mice with running activity that fell within the interquartile range were selected for further analyses.

### 2.3. Single FDB Fiber Isolation

All resting myoplasmic Ca^2+^, releasable SR Ca^2+^ store content, store-operated Ca^2+^ entry, constitutive Ca^2+^ entry, and electrically evoked Ca^2+^ transients studies were conducted using single, acutely dissociated *flexor digitorum brevis* (FDB) fibers, as previously described [57]. Briefly, FDB muscles were dissected from hind limb footpads and placed in Ringer’s solution (145 mM NaCl, 5 mM KCl, 2 mM CaCl_2_, 1 mM MgCl_2_, 10 mM HEPES, pH 7.4) supplemented with 1 mg/mL collagenase A (Roche Diagnostics, Indianapolis, IN, USA) while rocking gently at 37 °C for 1 h. Individual FDB fibers were then liberated on glass-bottom dishes by gentle trituration in Ringer’s solution using three sequentially increasing gauge glass pipettes. Fibers were then left undisturbed for at least 20 min to enable them to settle and stick to the bottom of the dish. Only healthy fibers with clear striations and no observable damage were used for experiments.

### 2.4. Resting Ca^2+^ Measurements

Resting free myoplasmic Ca^2+^ concentration was determined as previously described [57] Briefly, isolated FDB fibers were loaded with 4 µM fura-2 AM (Thermo Fisher, Carlsbad, CA, USA) in Ringer’s solution at room temperature (RT) for 30 min followed by a 30 min washout in dye-free Ringer’s solution. Fura-2 AM-loaded fibers were then placed on the stage of an inverted epifluorescence microscope (Nikon Instruments, Melville, NY, USA) and alternatively excited at 340 and 380 nm (20 ms exposure per wavelength, 2 × 2 binning) using a monochromator-based illumination system with fluorescence emission at 510 nm captured using a high-speed QE CCD camera (TILL Photonics, Graefelfing, Germany). 340/380 ratios from cytosolic areas of interest were calculated using TILL vision software (version 4.0), analyzed using ImageJ (version 1.54g, NIH) and converted to resting free Ca^2+^ concentrations using a fura-2 calibration curve approach described previously [58]. Biological replicates were plotted as the average of 2–12 technical replicates.

### 2.5. Measurements of Total Releasable Ca^2+^ Store Content

Total releasable Ca^2+^ store content was determined as previously described [57]. Briefly, FDB fibers were loaded with 5 µM fura-FF AM (AAT Bioquest, Sunnyvale, CA, USA), a low-affinity ratiometric Ca^2+^ dye, at RT for 30 min, followed by a 30 min washout in dye-free Ringer’s. Total releasable Ca^2+^ store content was calculated from the peak change in fura-FF ratio (ΔRatio_340/380_) upon application of a Ca^2+^ release cocktail (10 µM ionomycin, 30 µM cyclopiazonic acid, and 100 µM EGTA, referred to as ICE) in Ca^2+^-free Ringer’s solution. The peak change in fura-FF ratio was calculated using Clampfit 10.0 (Molecular Devices, Sunnyvale, CA, USA) and Microsoft Excel (version 16.99.2, Microsoft Corporation, Redmond, WA, USA). Biological replicates were plotted as the average of 2–7 technical replicates.

### 2.6. Measurements of Store-Operated and Constitutive Ca^2+^ Entry

Mn^2+^ quench experiments were used to quantify store-operated Ca^2+^ entry (SOCE, with store depletion) and constitutive Ca^2+^ entry (without store depletion) as previously described [57]. Briefly, FDB fibers were loaded with 5 μM fura-2 AM for 1 h at 37 °C in a Ca^2+^-free Ringer’s solution containing (in mM): 145 NaCl, 5 KCl, 1 MgCl_2_, 0.2 EGTA (pH 7.4). When measuring maximal SOCE activity, 1 μM thapsigargin and 15 μM cyclopiazonic acid (inhibitors of SERCA activity used to fully deplete SR Ca^2+^ stores), as well as 30 μM N-benzyl-p-toluene sulfonamide (BTS, a skeletal muscle myosin inhibitor to prevent movement artifacts [9] were included during fura-2 AM loading. In a second set of studies to assess constitutive Ca^2+^ entry, fibers were loaded with fura-2 AM and BTS in the absence of SERCA pump inhibitors. Both store-depleted and non-depleted fibers were then bathed in Ca^2+^-free Ringer’s and excited at 362 nm (isosbestic point of fura-2) while emission was detected at 510 nm using a DeltaRam illumination system (Photon Technologies Inc, Birmingham, NJ, USA). After obtaining an initial basal rate of fura-2 decay (R_baseline_), fibers were exposed to Ca^2+^-free Ringer’s supplemented with 0.5 mM MnCl_2_. The maximum rate of fura-2 quench in the presence of Mn^2+^ (R_max_) was determined from the peak differential of the fura-2 emission trace during Mn^2+^ application. The maximum rate of SOCE (R_SOCE_) was calculated as R_SOCE_ = R_max_ − R_baseline_ and expressed as dF/dt in counts/s [9]. Biological replicates (i.e., mice) were plotted as the average of 10 technical (i.e., individual fiber) replicates.

### 2.7. Measurements of Electrically Evoked Ca^2+^ Transients

Electrically evoked myoplasmic Ca^2+^ transients were monitored in singly isolated FDB fibers as described previously [57,59]. Briefly, FDB fibers were loaded with 4 µM mag-fluo-4 for 20 min at RT followed by washout in dye-free solution supplemented with 25 µM BTS for 20 min. While continuously being perfused with a control Ringer’s solution supplemented with 25 μM BTS, fibers were stimulated with a series of 5 electrically evoked twitch (1 Hz) stimulations followed by a single high frequency (500 ms at 100 Hz) train of stimulations using an extracellular electrode placed adjacent to the fiber of interest. Mag-fluo-4 was excited at 480  ±  15 nm using an Excite epifluorescence illumination system (Nikon Instruments, Melville, NY, USA) and fluorescence emission at 535  ±  30 nm was monitored with a 40× oil objective and a photomultiplier detection system (Photon Technologies Inc., Birmingham, NJ, USA). Relative changes in mag-fluo-4 fluorescence from baseline (F/F_0_) were recorded using Clampex 9.0 software (Molecular Devices, San Jose, CA, USA). The maximum rate of electrically evoked Ca^2+^ release was approximated from the peak of the first derivative of the mag-fluo-4 fluorescence (dF/dt) during electrical stimulation. The decay phase of each transient was fitted according to the following second order exponential equation:F(t) = A_fast_ × [exp(−t/τ_fast_)] + A_slow_ × [exp(−t/τ_slow_)](1)
where F(t) is the fluorescence at time t, A_fast_ and τ_fast_ are the amplitude and time constants of the fast component, respectively, and A_slow_ and τ_slow_ are the amplitude and time constants of the slow component, respectively. Biological replicates were plotted as an average of 2–9 technical replicates.

### 2.8. Ex Vivo Measurements of Muscle Contractile Function

Ex vivo assessment of muscle force production was made in intact excised EDL and *soleus* muscles attached to a servo motor and force transducer 1200 A (Aurora Scientific, Aurora, ON, Canada) and electrically stimulated using two platinum electrodes in a chamber continuously perfused with oxygenated Ringer’s solution at 30 °C as previously described [54,60,61,62]. Optimal stimulation and muscle length (L0) were determined using a series of 1 Hz twitch stimuli while stretching the muscle to a length that generated maximal force (F0). After establishing L0, muscles were equilibrated using three tetani (500 ms, 150 Hz) given at 1 min intervals and then subjected to a standard force frequency protocol (from 1 to 250 Hz). Muscle force was recorded using Dynamic Muscle Control software (version 5.500, Aurora Scientific, Aurora, ON, Canada) and analyzed using a combination of Dynamic Muscle Analysis (version 5.321, Aurora Scientific, Aurora, ON, Canada) and Clampfit 10.0 (Molecular Devices) software. Physiologic cross-sectional area (P-CSA) was calculated as:P-CSA = (muscle weight [mg])/(1.056 × (0.44 or 0.71) × length [mm])(2)
where 1.056 = muscle density [g/cm^3^], 0.44 = EDL angular factor, 0.71 = *soleus* angular factor [63].

### 2.9. Tissue Sectioning and Immunostaining

Muscles were removed, placed in 30% sucrose overnight at 4 °C, embedded in OCT (Tissue Tek, Sakura Finetek, Torrance, CA, USA), flash frozen using dry ice-cooled isopentane, stored at −80 °C and sectioned at 10 µm thickness. Prior to immunostaining, tissue sections were permeabilized with PBS-T (0.2% Triton-x-100 in PBS) for 10 min, blocked in 10% normal goat serum (NGS, Jackson ImmunoResearch, West Grove, PA, USA) for 30 min at RT, and then blocked in 3% AffiniPure Fab fragment goat anti-mouse (Jackson ImmunoResearch, West Grove, PA, USA)) with 2% NGS at RT for 1 h to decrease mouse antibody non-specific binding. Primary antibodies were applied in 2% NGS/PBS and then incubated for 2 h at RT or overnight at 4 °C followed by incubation with secondary antibody for 1 h at RT. All slides were mounted with Fluoromount-G (SouthernBiotech, Birmingham, AL, USA). Sections were imaged at 4× magnification on the Revolve (Echo, San Diego, CA, USA) microscope and analyzed using ImageJ (version 1.54g, NIH). Sample analyses were performed by investigators blinded to experimental group.

### 2.10. Antibodies

The following antibodies were used: rat anti-laminin-α2 (1:1500, L0663, Sigma-Aldrich, St. Louis, MO, USA), mouse anti-BA-D5 (MyHC-I, IgG2b, 1:40, Developmental Studies Hybridoma Bank (DSHB), Iowa City, IA, USA), mouse anti-SC-71 (MyHC-IIA, IgG1, 1:40, DSHB, Iowa City, IA, USA), mouse anti-BF-F3 (MyHC-IIB, IgM, 1:40, DSHB, Iowa City, IA, USA), mouse anti-Glyceraldehyde-3-phosphate dehydrogenase (GAPDH) (1:50,000, AM4300, Thermo Fisher, Carlsbad, CA, USA), mouse anti-CASQ1 (1:5000, MA3-913, Affinity BioReagents, Golden, CO, USA), rabbit anti-pan SERCA (1:10,000, sc-30110, Santa Cruz Biotechnology, Dallas, TX, USA), rabbit anti-voltage-dependent anion channel (VDAC) (1:5000, AB10527, Thermo Fisher, Carlsbad, CA, USA), rabbit anti-STIM1 (1:3000, S6197, Sigma Aldrich, St. Louis, MO, USA), rat anti-HA (1:3000, ROAHAHA Sigma-Aldrich, St. Louis, MO, USA,), rabbit anti-OPA1 (1:1000, D6U6N, Cell Signaling Technology, Danvers, MA, USA), rabbit anti-COL1A1 (1:1000, AB765P, EMD Millipore, Burlington, MA, USA), rabbit anti-ACTN2 (1:2000, 14221-1-AP, Proteintech, Rosemont, IL, USA), rabbit anti-CYP2E1 (1:2000, H00001571-D01P, Thermo Fisher, Carlsbad, CA, USA), rabbit anti-PPIF (1:2000, 45-5900, Thermo Fisher, Carlsbad, CA, USA), Alexa Fluor 405-conjugated goat anti-mouse IgG2b (1:1500, A-21141, Thermo Fisher, Carlsbad, CA, USA), Alexa Fluor 488-conjugated goat anti-mouse IgM (1:1500, A-21042, Thermo Fisher, Carlsbad, CA, USA), Alexa Fluor 594-conjugated goat anti-mouse IgG1 (1:1500, A-21125, Thermo Fisher, Carlsbad, CA, USA), AlexaFluor 647-conjugated goat anti-rat IgG (1:1500, A-21247, Thermo Fisher, Carlsbad, CA, USA), goat anti-rabbit IRDye800 (1:10,000, LiCor, Lincoln, NE, USA), goat anti-mouse IRDye800 (1:10,000, LiCor, Lincoln, NE, USA), goat anti-mouse IRDye700 (1:10,000, LiCor, Lincoln, NE, USA), and donkey anti-rat IgG-horseradish peroxidase conjugate (1:10,000, Jackson ImmunoResearch, West Grove, PA, USA).

### 2.11. Proteomic Sample Preparation

Whole *soleus* and EDL muscles from male mice were flash-frozen in liquid nitrogen and stored at −80 °C until ready for use. Samples were homogenized in 250 µL of 5% SDS, 100 mM TEAB and sonicated (QSonica, Newtown, CT, USA) for 5 cycles with 1 min incubation on ice after each cycle. Samples were then centrifuged at 15,000× *g* for 5 min to collect the supernatant. A bicinchoninic (BCA) assay (Thermo Fisher, Carlsbad, CA, USA) was used for protein quantitation. Samples were diluted to 1 mg/mL in 5% SDS, 50 mM TEAB. 25 µg of protein from each sample was reduced with dithiothreitol (2 mM) and incubated at 55 °C for 1 h. Iodoacetamide (10 mM) was then added and samples were incubated at RT for 30 min in the dark to alkylate proteins. Phosphoric acid (1.2%) was added, followed by six volumes of 90% methanol, 100 mM TEAB. Samples were added to S-Trap micros (Protifi, Fairport, NY, USA), and centrifuged at 4000× *g* for 1 min. S-Traps were washed twice by centrifuging through 90% methanol, 100 mM TEAB. 20 µL of 100 mM TEAB with 1 µg trypsin added to the S-Trap, followed by an additional 20 µL of TEAB. Samples were incubated at 37 °C overnight. S-Traps were centrifuged at 4000× *g* for 1 min to collect the digested peptides. Sequential additions of 0.1% trifluoroacetic acid (TFA) in 50% acetonitrile were added to the S-trap, centrifuged, and pooled. Samples were frozen and dried in a Speed Vac (Labconco, Kansas City, MO, USA) and resuspended in 0.1% TFA prior to mass spectrometry analysis.

### 2.12. Mass Spectrometry

Peptides were injected onto 30 cm C18 columns with 1.8 μm beads (Sepax, Newark, DE, USA) on an Easy nLC-1200 HPLC (Thermo Fisher, Carlsbad, CA, USA) connected to a Fusion Lumos Tribrid mass spectrometer (Thermo Fisher, Carlsbad, CA, USA) operating in data-independent mode. Ions were introduced to the mass spectrometer using a Nanospray Flex source operating at 2 kV. The gradient proceeded as 3% solvent B (0.1% formic acid in 80% acetonitrile) for 2 min, 10% solvent B for 6 min, 38% solvent B for 65 min, 90% solvent B for 5 min, held for 3 min, returned to starting conditions for 2 min and re-equilibrated for 7 min. The full MS1 scan was conducted from 395–1005 *m*/*z*, with a resolution of 60,000 at *m*/*z* of 200, an AGC target of 4 × 105 and a maximum injection time of 50 ms. MS2 scans were performed by using higher energy dissociation with a staggered windowing scheme of 14 *m*/*z* with 7 *m*/*z* overlaps, with fragment ions analyzed in the Orbitrap with a resolution of 15,000, an automatic gain control (AGC) target of 4 × 105 and a maximum injection time of 23 ms.

### 2.13. Proteomic Data Analyses

Raw data was processed using DIA-NN version 1.8.1 (https://github.com/vdemichev/DiaNN, accessed on 1 July 2022) in library-free analysis mode [64]. Library annotation was conducted using the mouse UniProt ‘one protein sequence per gene’ database (UP000000589_10090) with deep learning-based spectra and RT prediction enabled. For precursor ion generation, the maximum number of missed cleavages was set to 1, maximum number of variable modifications set to 1 for Ox(M), peptide length range set to 7–30, precursor charge range set to 2–3, precursor *m*/*z* range set to 400–1000, and fragment *m*/*z* range set to 200–2000. Quantification was set to ‘Robust LC (high precision)’ mode with cross-run normalization set to RT-dependent, MBR enabled, protein inferences set to ‘Genes’, and ‘Heuristic protein inference’ turned off. MS1 and MS2 mass tolerance and scan window size were automatically set by the software. Precursors were subsequently filtered at library precursor q-value (1%), library protein group q-value (1%), and posterior error probability (50%). Protein quantification was carried out using the MaxLFQ algorithm implemented in the DIA-NN R package. Peptide number was quantified in each protein group and implemented in the DiannReportGenerator package (https://github.com/URMC-MSRL/DiannReportGenerator, accessed on 1 July 2022) [65]. Only significantly different (*p* < 0.05) proteins within the top three quartiles of mean peptide abundance were included in analysis. Individual group comparison pathway and network analyses were conducted using ShinyGO 0.77 (South Dakota State University, Brookings, SD, USA) [66,67,68]. The top ten identified pathways were extracted from Gene Ontology (GO) Biological Process, GO Cellular Component, and KEGG pathway databases. The false discovery rate (FDR) cutoff was set at 0.05 and pathways were sorted by −Log_10_(FDR). Volcano plots were generated using the web-based software VolcaNoseR [69]. Volcano plot significance thresholds were set at −Log_10_ > 1.3 and the Log_2_ fold change threshold was set to exclude values between −1 and 1. Pie charts and individual pathway heatmaps were generated using Microsoft Excel (version 16.99.2, Microsoft Corporation, Redmond, WA, USA).

### 2.14. Electron Microscopy and Histology

Intact EDL and *soleus* muscles were fixed at RT in 0.1 M sodium cacodylate (NaCaCO)-buffered 3.5% glutaraldehyde solution (pH 7.2) and processed as previously described [70,71,72]. Briefly, fixed muscles were post-fixed in 2% OsO_4_ for 1–2 h, rinsed with 0.1 M NaCaCO buffer, en bloc stained with saturated uranyl acetate replacement, and embedded for electron microscopy (EM) in epoxy resin (Epon 812). Semithin (700 nm) and ultrathin sections (~40 nm) were cut on a Leica Ultracut R microtome (Leica Microsystem, Austria) using a Diatome diamond knife (DiatomeLtd. CH-2501, Biel, Switzerland). For histological examination, 700 nm thick sections were stained in a solution containing 1% toludine blue O and 1% sodium tetraborate in ddH_2_O for 3 min on a hot plate at 55–60 °C. After washing and drying, sections were mounted with DPX mounting medium (Sigma Aldrich, St. Louis, MO, USA) for histology and observed with a Leica DMLB light microscope (Leica Microsystem, Vienna, Austria) connected to a Leica DFC450 camera equipped with Leica Application Suite v4.6 for Windows (Leica Microsystem, Vienna, Austria). For EM, ultrathin sections (~40 nm) were cut using a Leica Ultracut R microtome (Leica Microsystems, Vienna, Austria) with a Diatome diamond knife (Diatome, Biel, Switzerland) and double-stained with uranyl acetate replacement and lead citrate. Sections were viewed on a FP 505 Morgagni Series 268D electron microscope (FEI Company, Brno, Czech Republic), equipped with a Megaview III digital camera (Olympus Soft Imaging Solutions, Munster, Germany) and Soft Imaging System at 60 kV. For all quantitative EM analyses, micrographs of non-overlapping regions were randomly collected from transversal and longitudinal sections of internal areas of fibers. The percentage of fibers containing TAs, the number of TAs per fiber, and the average size of TAs (μm^2^) were evaluated in transversal sections of EDL muscles at low-medium magnification and were reported as an average per biological replicate. The total number of mitochondrial profiles, mitochondrial volume, and frequency of altered mitochondria were evaluated in longitudinal ultrathin sections of *soleus* muscles at 8900–14,000× magnification and reported as average number per 100 μm^2^. Mitochondria were classified as altered when: (a) the external membrane was disrupted, (b) internal cristae were severely vacuolated, and/or (c) contained myelin figures. The number of disordered myofibrils (i.e., presenting one or more sarcomeric unit with misalignment of the Z line with respect to the upper and lower myofibrils) was counted in micrographs taken from longitudinal sections of *soleus* muscles at 8900–14,000× magnification and reported as average number per 100 μm^2^.

### 2.15. Western Blot Analyses

EDL and *soleus* muscles were flash frozen in liquid nitrogen and stored at −80 °C until ready for use. Muscles were mechanically homogenized in RIPA lysis buffer (20 mM Tris-HCl pH 7.5, 150 mM NaCl, 1 mM Na_2_EDTA, 1 mM EGTA, 1% NP-40, 1% sodium deoxycholate, 1 mM Na_3_VO_4_, 10 mM NaF) supplemented with Halt protease inhibitor, as recommended by the manufacturer. Samples were centrifuged at 13,000× *g* for 30 min; supernatants were retained and protein concentration was determined using the Bio-Rad DC assay (500-0116). A total of 5 µg of total protein was separated on 12% sodium dodecyl sulfate-polyacrylamide gel electrophoresis and transferred to a nitrocellulose membrane. Membranes were briefly stained with a 0.1% Ponceau S solution (Sigma Aldrich, St. Louis, MO, USA, P3504) to ensure equal protein loading and for normalization of target protein expression. Membranes were blocked in non-fat dry milk (1–5%) or bovine serum albumin (3%)-supplemented TBS-T (20 mM Tris, 150 mM NaCl, 0.1% Tween 20, pH 7.4) for 1 h and probed overnight at 4 °C while shaking with primary antibodies diluted in TBS-T. Membranes were washed 3x in TBS-T and secondary antibodies were applied, diluted in TBS-T supplemented with 1–5% non-fat dry milk for 1 h at RT while shaking. Blots were imaged on a LI-COR Odyssey (LI-COR Biosciences, Lincoln, NE, USA) gel imaging system or KwikQuant Pro Imager (Kindle Biosciences, Greenwich, CT, USA) with band intensity quantified using Image Studio Lite (LI-COR Biosciences, Lincoln, NE, USA). For HA blot quantitation, membranes were incubated for 2 min RT shaking in KwikQuant Ultra HRP Substrate Solution A and B (1:1), diluted 1:5 in ddH_2_O before imaging for 1–240 s exposure time.

### 2.16. Statistical Analyses

All statistical calculations were performed using GraphPad Prism 9 software. Statistical significance was determined through Student’s t-tests with Welch’s corrections (unpaired, two-tailed, 95% confidence interval) or ANOVA (two-way, followed by Tukey multiple comparisons test), where *p* < 0.05 was considered statistically significant (*/#/&/$ *p* < 0.05, **/##/&&/$$ *p* < 0.01, ***/###/&&&/$$$ *p* < 0.001, ****/####/&&&&/$$$$ *p* < 0.0001). Isolated */#/&/$ represent ANOVA group effects or interaction of variables. Error bars are represented as +/− standard error of the mean (S.E.M.).

## 3. Results

### 3.1. WT and GS Mice Display Similar Voluntary Wheel Running Activity and Body Mass Change Following Six Months of Voluntary Wheel Running

Humans harboring the *ORAI^G98S/+^* mutation typically experience a slowly progressive myopathy with onset of symptoms commonly reported in childhood [26,43]. Zhao et al. showed that myotubes and FDB fibers from young (≤five weeks old) *Orai1^G100S/+^* (GS) mice exhibit increased constitutive Ca^2+^ entry in the absence of TAs, whereas FDB fibers from eight-month-old adult GS mice display TAs but lack both constitutive and store-operated Ca^2+^ entry [50]. Since sustained VWR (from 9–24 months of age) restored SOCE and reduced the development of TAs in aged WT male mice [53], we singly housed WT and GS mice in cages with either free-spinning (VWR) or locked (sedentary control) wheels from two to eight months of age (Figure 1A). While myalgia and exercise-induced cramps are associated with the presence of TAs [73,74,75,76], average daily running activity was not significantly different between eight-month-old WT and GS mice (Figure 1B). At two months of age prior to VWR, no difference in body mass was observed between WT and GS mice (Figure 1C). However, at eight months of age, GS mouse body mass was significantly lower than age-matched WT mice both with and without VWR (Figure 1D). Furthermore, VWR from two to eight months of age significantly reduced body mass gain irrespective of genotype (Figure 1E).

### 3.2. Voluntary Wheel Running Normalizes Total Releasable Ca^2+^ Store Content in the Absence of Restoration of Store-Operated Ca^2+^ Entry in FDB Fibers from GS Mice

Myotubes derived from TAM patients exhibit increased Ca^2+^ entry and elevated releasable Ca^2+^ store content [17,26,27,29,41,43]. However, FDB fibers from GS mice exhibit increased constitutive Ca^2+^ entry only early in development (≤five weeks of age) and SOCE is reduced across all ages tested (myotubes–18 months of age) [50]. While no differences in resting myoplasmic Ca^2+^ concentration were observed in the current study between any of the four cohorts at eight months of age (Figure 2A), total releasable Ca^2+^ store content was increased in FDB fibers from sedentary GS mice compared to that of sedentary WT mice and this difference was normalized to that of fibers from WT mice following VWR (Figure 2B). As sustained VWR was shown previously to both restore SOCE and reduce TAs in aged male mice [57], we assessed the impact of sustained voluntary endurance exercise on constitutive/store-operated Ca^2+^ entry. Following six months of VWR (two to eight months of age), constitutive and store-operated Ca^2+^ entry was quantified in FDB fibers from eight-month-old WT and GS mice using Mn^2+^ quench of fura-2 fluorescence as a surrogate measure of divalent cation entry. While SOCE was significantly increased in WT mice after six months of VWR, SOCE remained similarly reduced in FDB fibers isolated from both sedentary and VWR GS mice (Figure 2C,D). Similarly, constitutive Ca^2+^ entry was significantly increased following VWR in FDB fibers from WT mice, but not GS mice (Figure 2E,F). Consistent with reduced constitutive and store-operated Ca^2+^ entry in eight-month-old GS mice, ORAI1 expression was significantly reduced in both *soleus* (Appendix A) and, to a greater extent, EDL (Appendix A) muscle of GS mice. Consistent with the selective increase in SOCE in FDB fibers from WT mice following sustained voluntary endurance exercise, STIM1 expression was significantly increased in EDL (Appendix A), but not *soleus* (Appendix A), muscle following six months of VWR.

The magnitude and kinetics of electrically evoked Ca^2+^ transients were assessed using the rapid, low-affinity Ca^2+^ dye mag-fluo-4. Peak electrically evoked Ca^2+^ release during a single twitch stimulation was significantly reduced in FDB fibers from GS mice following six months of VWR compared to that observed for fibers from either sedentary GS mice or WT mice following VWR (Appendix A). Although six months of VWR prolonged the slow component (τ_slow_) of Ca^2+^ transient decay, which primarily reflects the rate of SERCA-mediated SR Ca^2+^ reuptake (Appendix A) and is consistent with the reduced expression of SERCA in EDL muscle from both WT and GS exercised mice (Appendix A), all other kinetic parameters of Ca^2+^ transient increase and decay were not significantly different across the four experimental cohorts (Appendix A).

### 3.3. Voluntary Wheel Running Improves Ex Vivo Contractile Function of Soleus Muscles from GS and WT Mice

Both TAM patients and TAM mouse models display progressive muscle weakness [17,18,22,23,26,28,29,32,43,48,77]. Thus, we assessed EDL and *soleus* muscle mass and ex vivo contractile function. Raw EDL mass was not different between sedentary WT and GS mice nor was it significantly altered for either genotype following six months of VWR (Figure 3A). However, VWR did result in a modest increase in GS EDL muscle mass normalized as a proportion of total body mass (Figure 3B), presumably due to the reduced body mass observed in these mice following VWR (Figure 1E). Absolute force elicited via maximal ex vivo stimulation was significantly reduced in EDL muscles from sedentary GS mice compared to that of sedentary WT mice, with a significant genotype group effect observed regardless of exercise (Figure 3C,D). EDL specific force was similarly reduced in GS mice compared to that of WT mice regardless of exercise (Figure 3E,F). The maximal rates of EDL force production (Figure 3G) and relaxation (Figure 3H) were also significantly reduced in GS mice, as reflected in a significant group effect of genotype. EDL muscles from GS mice exhibited a modest, but significant, reduction in type IIb fibers and a corresponding increase in type IIx fibers (Appendix A). Average EDL fiber size was unaltered for all fiber types across all four conditions (Appendix A).

In the *soleus*, significant group effects of genotype (GS reduced compared to WT) and VWR (exercise increased compared to sedentary) were observed for raw muscle mass (Figure 3I). Normalized *soleus* mass (as a proportion of total body mass) was significantly increased in both WT and GS mice following VWR (Figure 3J). Importantly, absolute (Figure 3K,L) and specific (Figure 3M, N) force were significantly increased in *soleus* muscles from GS mice following six months of VWR. In fact, a group effect of VWR on absolute and specific force was observed regardless of genotype. Moreover, maximal rates of *soleus* force production (Figure 3O) and relaxation (Figure 3P) were significantly increased following VWR with a greater proportional rescue of relaxation rate observed for GS mice following VWR. Despite this effect of VWR on accelerating the rate of *soleus* contractile force generation/relaxation, the proportion of fast type IIb and IIx fibers was reduced (Appendix A) and type IIa fiber cross-sectional area (CSA) was increased (Appendix A) in GS mice following VWR.

### 3.4. Voluntary Wheel Running Reduces the Number of Structurally Altered Mitochondria and Improves the Alignment of Sarcomeres in Soleus Muscle of GS Mice

Conditions exhibiting impaired intracellular Ca^2+^ regulation in skeletal muscle often display a mitochondrial phenotype, as mitochondrial function is regulated by mitochondrial Ca^2+^ uptake and accumulation [78,79]. Slow twitch oxidative muscle (i.e., *soleus*) relies heavily on proper mitochondrial function for its energetic demands, as slow twitch muscle is comprised primarily of mitochondria-rich type I and type IIa fibers (Appendix A). Thus, *soleus* muscles from sedentary and VWR, WT and GS mice were fixed for histological and electron microscopic analyses (Figure 4A–F). Mitochondrial density and volume were similar in *soleus* muscles across all cohorts (Figure 4G,H). Consistent with this, VDAC expression was not significantly different between *soleus* muscles from sedentary and VWR, GS or WT mice (Appendix A). However, the percentage of structurally altered mitochondria (identified as containing either disrupted external membranes, severely vacuolated internal cristae, and/or containing myelin figures) was significantly higher in sedentary GS mice compared to sedentary WT mice, while GS VWR mice displayed no significant difference compared to WT Sed mice (Figure 4I). Interestingly, OPA1, a protein involved in inner mitochondrial membrane fusion, cristae organization, and SR-mitochondrial Ca^2+^ signaling [80], was upregulated in *soleus* muscle of both WT and GS mice following VWR (Appendix A). Additionally, *soleus* sarcomere misalignment was more prevalent in sedentary GS mice compared to sedentary WT mice, while no significant difference was observed between sedentary WT mice and GS mice following VWR (Figure 4J).

### 3.5. Alterations in the Soleus Muscle Proteome from GS Mice Are Normalized Following Voluntary Wheel Running

Proteomic analyses were conducted to provide a comprehensive assessment of the proteins and pathways that are altered in the *soleus* and EDL muscles of sedentary eight-month-old WT mice. These studies revealed 1270 significantly altered proteins between the two groups (Appendix A). As expected, significantly altered pathways included metabolism, mitochondria, and pathways encompassing muscle fiber type (myofibril, contractile fiber), as was expected (Appendix A).

Comparative proteomic studies were also conducted on *soleus* muscles from 8-month-old sedentary WT and GS mice. A total of 149 proteins (out of 2124 proteins) were either significantly upregulated (81 proteins) or downregulated (68 proteins) in *soleus* muscle of eight-month-old sedentary GS mice compared to that of age-matched sedentary WT mice (Figure 5A,B). In contrast, only 96 proteins (18 upregulated and 78 downregulated), were significantly altered in the *soleus* muscle of GS mice following six months of VWR, as compared to WT VWR samples (Figure 5E,F). The proteomics results in *soleus* muscles from sedentary eight-month-old WT and GS mice were validated by quantitative Western blot analyses. These studies confirmed changes in proteins identified by proteomic analysis to be significantly upregulated (CYP2E1) and downregulated (PPIF) in *soleus* muscle of GS mice (Appendix A). Further, VDAC and STIM1 proteins identified as being unchanged by proteomic analysis of *soleus* muscle were also confirmed as being unaltered by Western blot analysis (Appendix A).

Interestingly, all but two of the significantly upregulated proteins and two of the significantly downregulated proteins observed in *soleus* muscle of sedentary GS mice were corrected after VWR. This correction of an altered *soleus* muscle proteome is reflected in the narrowing of the volcano plot of all identified proteins following six months of VWR (compare Figure 5B,F). GO Cellular process, GO Biological process, and KEGG pathway and network analyses were conducted to identify key disease hallmarks in *soleus* muscle of GS mice and mitigation with sustained voluntary endurance exercise (Appendix A). The top 10 GO Cellular process terms involved changes in mitochondria, intracellular organelles, and myofibrils (Figure 5C). Network analysis of these terms revealed a tight system of interactions between these pathways (Figure 5D). While “Mitochondrion” was the most significantly altered Cellular process identified in *soleus* muscles of both sedentary and VWR GS and WT mice, most of the other GO Cellular process terms identified in *soleus* of sedentary GS mice were absent after VWR (Figure 5G). In addition, the tight mitochondrial/organelle/myofibril network of proteins observed in *soleus* muscle of sedentary GS mice was disrupted following six months of VWR (compare Figure 5D,H). Importantly, assessment of all 158 proteins within the Mitochondrion GO Cellular process term identified under both sedentary and VWR conditions revealed a normalization of most up- and down-regulated proteins following six months of VWR (compare heat maps in Figure 5I and Appendix A). In agreement with mitochondrial alterations being a major point of convergence for pathology and prevention, the top identified Biological and KEGG pathways between *soleus* muscle of sedentary GS and WT mice centered around alterations in fatty acid and aerobic metabolism (Appendix A). Following six months of VWR, pathways associated with protein translation and biosynthetic processes were among the top altered pathways identified between both genotypes (Appendix A) and within the same genotype (Appendix A).

### 3.6. Voluntary Wheel Running Reduces TA Prevalence and Size in EDL Muscles of GS Mice

As mentioned previously, TAs are more frequently observed in type II glycolytic fibers of fast twitch muscle in both humans with TAM and aged male mice that develop TAs spontaneously [39,40,42,81]. While studies in patients with TAM are lacking, we demonstrated that sustained voluntary endurance exercise greatly reduced TA formation in aged male mice [53]. Thus, we determined the impact of sustained voluntary endurance exercise (six months of VWR) on the prevalence of TAs observed in fast twitch EDL muscle of eight-month-old GS mice (Figure 6A–C). We found that EDL muscle from sedentary GS mice exhibited an increased percentage of fibers with TAs that was significantly reduced following six months of VWR (Figure 6D). In addition, the number of TAs per fiber and average TA size were both significantly higher in EDL muscles from sedentary GS mice compared to age-matched WT mice, while neither metric significantly differed between sedentary WT mice and GS mice after six months of VWR (Figure 6E,F). Several previous studies demonstrated that TAs are positive for SR markers, including CASQ1 and SERCA [26,29,39,40,43]. Furthermore, it is theorized that the highly ordered arrangement of TAs observed in aged male mice is in part due to “polymerization of SERCA into a semicrystalline arrangement” [39]. Consistent with the observed reduction in TAs following VWR (Figure 6), SERCA and CASQ1 expression were both significantly reduced in EDL muscle from GS VWR mice compared to GS sedentary mice (Appendix A).

### 3.7. The EDL Proteome from GS and Exercised Mice Is Significantly Altered and Converges on Pathways of Fiber Contractility and Supramolecular Complex Formation

Proteomic analyses were also conducted in EDL muscles from eight-month-old GS and WT mice under either sedentary or VWR conditions for six months. A total of 280 proteins (out of 1935 proteins) were either significantly upregulated (176 proteins) or downregulated (104 proteins) in EDL muscle of eight-month-old sedentary GS mice compared to that of age-matched sedentary WT mice (Figure 7A,B). The top 10 GO Cellular processes identified to be altered in EDL muscle of sedentary GS mice were primarily related to contractile fiber/myofibril/sarcomere and supramolecular complex/polymer/fiber pathways (Figure 7C), with a highly interconnected network between these pathways (Figure 7C,D). Interestingly, pathways implicating alterations in muscle innervation (e.g., synapse and myelin sheath) were also identified, though these pathways were outside the tight myofibril/supramolecular complex pathway network. Furthermore, KEGG pathway analysis of EDL muscles from eight-month-old sedentary GS and WT mice were consistent with changes observed in several disorders related to protein processing and neurodegeneration including Prion disease, Alzheimer disease, and Parkinson disease (Appendix A). Proteomics results in EDL muscles from sedentary eight-month-old WT and GS mice were also validated by quantitative Western blot analyses. These studies confirmed changes in proteins identified by proteomic analysis to be significantly upregulated (ACTN2) or downregulated (COL1A1) in EDL muscle of sedentary eight-month-old GS mice compared to age-matched WT mice (Appendix A). In addition, CASQ1 was confirmed by both proteomic and Western blot analyses to be significantly reduced following VWR in EDL muscle from GS mice (Appendix A).

Unlike proteomic analyses of *soleus* muscles, the comparison between EDL muscle of WT and GS mice following six months of VWR revealed over twice the number of significantly altered proteins compared to that observed under sedentary conditions. Specifically, a total of 573 proteins (125 downregulated and 448 upregulated) were significantly altered in the EDL muscle of GS mice following six months of VWR (Figure 7E). Consistent with the lack of a statistically significant rescue of EDL contractile force production (Figure 3F), both contractile fiber and myofibril Cellular process pathways remained significantly altered in EDL muscles of GS mice after six months of VWR (Figure 7G,H). Interestingly, while changes in mitochondrial/organelle related pathways were not identified in EDL muscles of sedentary GS mice (Figure 7C), these pathways represented the top five terms found to be significantly altered in EDL muscles of GS mice compared to that of WT mice after six months of VWR (Figure 7G,H, Appendix A) and were not identified in any of the other EDL comparisons (Appendix A). Importantly, supramolecular complex/polymer/fiber pathways were no longer identified as an altered GO Cellular process in EDL muscles from GS mice after VWR (Figure 7G,H). This observation is consistent with the reduction in TA prevalence observed in EDL muscles of eight-month-old GS mice following sustained voluntary endurance exercise (Figure 6). Additionally, cellular protein/macromolecule localization processes were within the top altered GO Biological pathways in EDL muscles from sedentary GS mice, which are interconnected with pathways involved in regulating intracellular transport (Appendix A). Top GO Biological processes identified when comparing EDL muscles from GS and WT mice after VWR include energy derivation by oxidation of organic compounds, oxidative phosphorylation, cellular/aerobic respiration, and cellular protein-containing complex assembly (Appendix A). Overall, proteomic results from EDL muscles of sedentary and VWR, WT and GS mice (Figure 7, Appendix A) support the notion that proteins involved in intracellular organelles and supramolecular complexes contribute to the formation of TAs and that sustained voluntary endurance exercise reduces TA prevalence in part by mitigating these effects.

## 4. Discussion

In the present manuscript we tested the impact of sustained voluntary endurance exercise (six months of VWR) on the skeletal muscle phenotype of *Orai1^G100S/+^* mice. Zhao et al. found that eight-month-old GS mice exhibit several key hallmarks of TAM in humans with the analogous G98S mutation in ORAI1 including progressive muscle weakness, elevated levels of serum creatine kinase, exercise intolerance and the histological presence of TAs in skeletal muscle fibers [50]. Importantly, we found a beneficial effect of six months of endurance exercise in reducing TAs in the fast twitch EDL muscle (Figure 6D–F), as well as improving contractile function (Figure 3K–P), limiting mitochondrial structural alterations, and preventing sarcomere misalignment (Figure 4I,J) in *soleus* muscle.

Individuals with TAM commonly experience exercise intolerance [28,29]. Consistent with this, eight-month-old GS mice exhibited more rests during forced treadmill running and falls during forced rotarod exercise [50], but no significant difference in average VWR activity was observed between WT and GS mice (Figure 1B). The reason for this apparent difference is not entirely clear but may reflect the lower overall body mass of GS mice (Figure 1D), the type of activity involved (forced acute fatiguing exercise paradigms versus voluntary endurance exercise), or the fact that VWR was initiated in mice at an earlier age (two months) while acute treadmill/rotarod challenges were initiated at older ages (i.e., at eight months of age) [13,82].

SOCE plays a key role in regulating multiple key skeletal muscle processes including muscle development [8,10,11] and fatigue resistance [9,12,13,14]. Conversely, disruptions in SOCE activity contribute to a wide range of muscle disease pathology and dysfunction including muscular dystrophy [82,83,84,85,86] and dynapenia [15,87,88]. One hour of acute treadmill exercise increases STIM1-ORAI1 co-localization, as well as both constitutive and store-operated Ca^2+^ entry through the formation of Ca^2+^ entry units within the I band region of the sarcomere that consist of junctions formed by transverse tubule extensions that interact with flat/parallel stacks of SR cisternae [57,89]. Similarly, we observed an increase in constitutive and store-operated Ca^2+^ entry (Figure 2B,D) in FDB fibers of WT mice after six months of VWR. However, no such increase in either constitutive or store-operated Ca^2+^ entry was observed in FDB fibers from GS mice after six months of VWR. The precise mechanism for why ORAI1 function was not enhanced in GS mice after sustained voluntary endurance exercise as is observed in WT mice is unclear but appears to be due at least in part to markedly reduced ORAI1 expression in skeletal muscle of GS mice under both sedentary conditions and after six months of VWR (Appendix A). Nevertheless, our findings are consistent with prior results [53] that found sustained voluntary endurance exercise protected against age-related skeletal muscle decline by enhancing SOCE, at least in the absence of TAM.

Since TAM disproportionately impacts fast twitch fibers [26,43,47,81,90], the identification of TAs in EDL, but not *soleus*, muscle fibers is not surprising. However, while TAM patients often present with type I fiber predominance and type II fiber hypotrophy/atrophy [26,27,30,43,90], this was not observed in eight-month-old GS mice. Despite robust mitigation of TA prevalence in the EDL muscle following six months of VWR, the functional deficit in EDL maximal specific force production was not increased with sustained VWR exercise. VWR typically promotes fiber type transitions from type IIb and IIx fibers toward more oxidative type IIa fibers [54,91,92]. In the *soleus*, we indeed observed an exercise-dependent reduction in type IIb and IIx fibers and parallel increase in type IIa fiber CSA (Appendix A). In the EDL, we observed a modest increase in type IIx fibers following six months of VWR (Appendix A). Regardless, given the structural (e.g., TAs) and functional (e.g., specific force deficit) impact of the G98S TAM mutation in ORAI1 on skeletal muscle, treatment in humans may best be addressed through a combination of endurance exercise in conjunction with resistance exercise or high-intensity interval training that are better suited to increase muscle hypertrophy and strength, particularly in type II fibers [93,94,95,96,97]. Also, our study highlights the possibility that interventional exercise may be most effective if implemented earlier in life before TAs are formed. Since exercise intolerance is a common symptom of TAM, future investigation of the effectiveness of exerkines, or ‘exercise-in-a-pill,’ in TAM is warranted.

The mitigation of TAs in the EDLs of GS mice with six months of VWR is robust and strikingly similar to the protection against age-associated TA formation in the EDLs of male mice following 15 months of VWR [53]. We further found that sustained VWR activity normalized both enhanced SERCA/CASQ1 expression (Appendix A) and total releasable Ca^2+^ store content (Figure 2F) observed in skeletal muscle from sedentary GS mice. These findings support prior proposals that TAs serve as a reservoir to sequester excess Ca^2+^ from the myoplasm to protect fibers from the deleterious effects of high concentrations of myoplasmic Ca^2+^ (Figure 2E) [17,26,29,40,41].

The improvement of *soleus* muscle function in both WT and GS mice after six months of VWR highlights a key central adaptation of skeletal muscle to endurance exercise: improved mitochondrial function/dynamics. Similarly to observations in different contexts of skeletal muscle decline such as myotonic dystrophy type 1 [98] and juvenile irradiation [54], prolonged endurance exercise improves muscle function in part due to favorable mitochondrial adaptations despite mitochondrial disruption not always being the primary mechanism driving muscle dysfunction. However, we previously reported proteomic alterations in mitochondrial pathways and significantly reduced mitochondrial function in skeletal muscle of GS mice [50], which we further confirmed in this study. An important distinction elucidated from this study is that a change in mitochondrial content could not explain the favorable exercise-induced mitochondrial adaptations observed in *soleus* muscle of GS mice (Figure 4G, H, Appendix A). Alternatively, we found that prolonged endurance exercise reduced the percentage of structurally altered mitochondria (Figure 4I), increased OPA1 expression (Appendix A), and normalized proteins within the Mitochondrion GO Cellular process (Figure 5I, Appendix A). Together, these changes indicate that six months of VWR improves mitochondrial quality and function in the absence of a change in overall mitochondrial content (Figure 3I–P).

While the findings in this study highlight key pathways and networks altered in muscle of GS mice that are likely to contribute to disease pathology, future studies are needed to build upon these findings and identify specific protein changes responsible for TAs and mitochondrial dysfunction, and to directly assess the impact of endurance training on mitochondrial function in skeletal muscle of GS mice. Another unaddressed aspect in this study of the myopathy in GS mice is the potential contribution of altered innervation and disruptions of the neuromuscular junction. Interestingly, several of the top altered pathways identified in the proteomic analyses of EDL muscles from sedentary WT and GS mice include neuron-related processes such as synapse, myelin sheath, and pathways involving neurodegeneration and neurodegenerative diseases (Appendix A).

It is known that males display more pronounced TAM symptoms as compared to females; however, females are also impacted by the disease. Thus, we included female mice in our analyses. Future studies would benefit from larger experimental cohorts in order to address sex-specific differences in disease pathology and benefit from exercise. Additionally, it will be of interest for future studies to determine whether GS mice exhibit some of the multisystemic aspects of Stormorken syndrome such as miosis, hyposplenism, ichthyosis, dyslexia, and thrombocytopenia as well as if prolonged endurance exercise is able to mitigate these features. Though GS mice exhibit normal levels of platelets and do not exhibit excess bleeding [50], VM mice exhibit these phenotypes, and thus, could be used to address this question.

## 5. Conclusions

In conclusion, our findings suggest that formation of TAs may be the result of reduced muscle activity. We provide evidence for the mitigation of TAs and muscle weakness with sustained voluntary endurance exercise and potential mechanistic avenues for future investigation that may contribute to TA formation/mitigation (proteins involved in the formation of supramolecular complexes) and muscle dysfunction (proteins involved in controlling mitochondrial function) in TAM.

## Figures and Tables

**Figure 1 cells-14-01383-f001:**
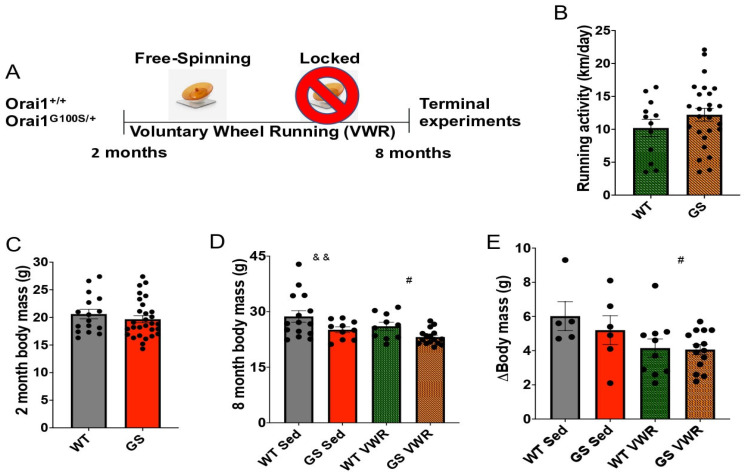
Experimental design and characterization of WT and GS mouse running activity and body mass following six months of VWR. (**A**) Timeline of six months VWR from two months to eight months of age in WT and GS mice. Mice were sacrificed at eight months of age to conduct terminal experiments. (**B**) Average daily running activity of WT and GS mice during six months of VWR. (**C**) WT and GS mouse body mass at two months of age. (**D**) WT and GS, Sed and VWR mouse body mass at eight months of age. (**E**) Change in body mass from two to eight months of age in WT and GS, Sed and VWR mice. Each dot is representative of one mouse. Two-way ANOVA # *p* < 0.05, && *p* < 0.01. Isolated &, # denote ANOVA group effects of genotype and exercise, respectively. Data displayed as mean +/− S.E.M.

**Figure 2 cells-14-01383-f002:**
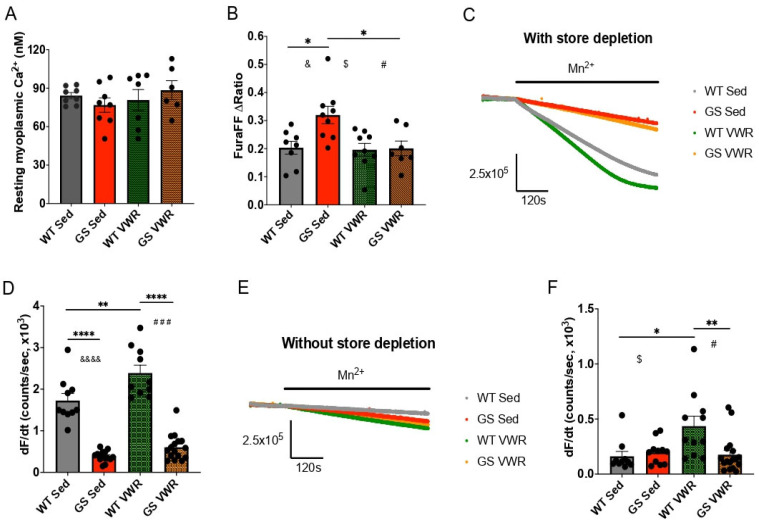
Ca^2+^ measurements from isolated FDB fibers. (**A**) Resting myoplasmic Ca^2+^ measurements of fibers from WT and GS, Sed and VWR mice. (**B**) Total releasable Ca^2+^ store content measurements in fibers from WT and GS, Sed and VWR mice. (**C**) Representative traces from Mn^2+^ quench assay with SR Ca^2+^ store depletion to assess SOCE in fibers from WT and GS, Sed and VWR mice. (**D**) Quantification of SOCE via rate of Mn^2+^ quench with SR Ca^2+^ store depletion from (**C**). (**E**) Mn^2+^ quench assay without SR Ca^2+^ store depletion to assess constitutive Ca^2+^ entry in WT and GS, Sed and VWR mice. (**F**) Quantification of constitutive Ca^2+^ entry via rate of Mn^2+^ quench without SR Ca^2+^ store depletion from (**E**). Each dot is representative of the mean from one mouse. Two-way ANOVA with multiple comparisons */$/#/& *p* < 0.05, ** *p* < 0.01, ### *p* < 0.001, ****/&&&& *p* < 0.0001. Isolated &, #, $ denote ANOVA group effects of genotype, exercise, and significant interaction of exercise and genotype, respectively. Data displayed as mean +/− S.E.M.

**Figure 3 cells-14-01383-f003:**
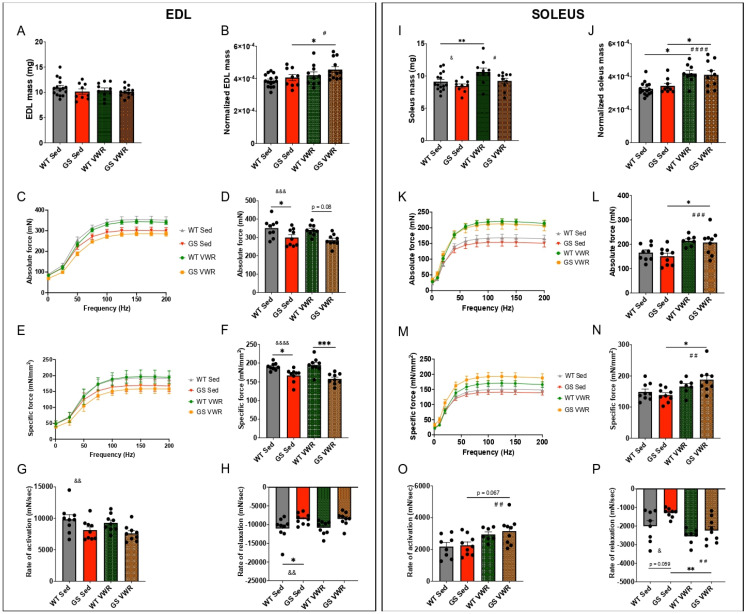
EDL (left) and *soleus* (right) muscle mass and ex vivo physiology upon electrical stimulation. Muscle mass and ex vivo physiology of WT and GS, Sed and VWR, EDL and *soleus* muscles. (**A**,**I**) Raw excised muscle mass. (**B**,**J**) Lean muscle mass calculated as raw muscle mass divided by body mass. (**C**,**K**) Absolute force frequency curve. (**D**,**L**) Quantification from (**C**,**K**) of maximal absolute force at 200 Hz stimulation. (**E**,**M**) Specific force frequency curve as normalized to muscle physiologic cross-sectional area. (**F**,**N**) Quantification from (**E**,**M**) of maximal specific force at 200 Hz stimulation. (**G**,**O**) Rate of activation measurements of excited muscle time to peak muscle contraction. (**H**,**P**) Rate of relaxation measurements of time to baseline following muscle excitation cessation. Each dot is representative of one mouse. Two-way ANOVA with multiple comparisons. */#/& *p* < 0.05, **/##/&& *p* < 0.01, ***/###/&&& *p* < 0.001, ####/&&&& *p* < 0.0001. Isolated &, # denote ANOVA group effects of genotype and exercise, respectively. Data displayed as mean +/− S.E.M.

**Figure 4 cells-14-01383-f004:**
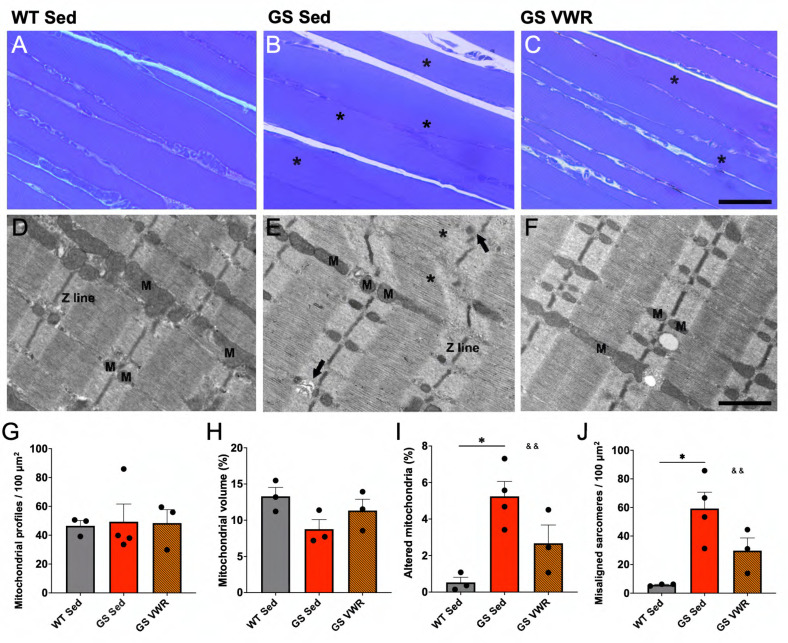
Assessment of mitochondrial structure and sarcomere alignment. (**A**–**C**) Histological and (**D**–**F**) EM images of longitudinal sections of *soleus* muscle from WT Sed (**A**,**D**), GS Sed (**B**,**E**), and GS VWR (**C**,**F**) mice. Asterisks in (**B**,**C**) point to fibers with non-regular pale-dark striations. In (**E**), asterisks indicate sarcomeres with Z line streaming and misaligned myofibrils; black arrows point to areas with partial disruption of the regular arrangement of the I band within a sarcomere. M indicates mitochondrial profiles. (**G**) Quantitative EM analysis of frequency of mitochondrial profiles per 100 µm^2^. (**H**) Quantitative EM analysis of mitochondrial volume as a percent of total myofiber volume analyzed. (**I**) Percentage of mitochondria profiles displaying altered structure. (**J**) Number of misaligned sarcomeres per 100 µm^2^. Each dot represents a single mouse. (**A**–**C**), scale bar = 200 µm; (**D**–**F**), scale bar = 1 µm Two-way ANOVA with multiple comparisons. * *p* < 0.05, && *p* < 0.01. Isolated & denotes ANOVA group effect of genotype. Data displayed as mean +/− S.E.M.

**Figure 5 cells-14-01383-f005:**
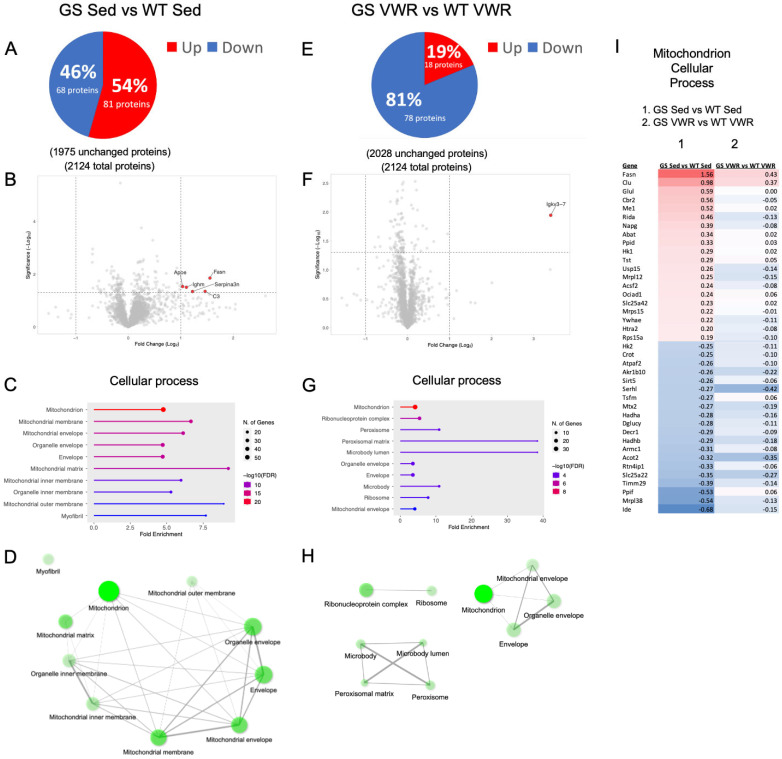
*Soleus* proteomic analysis of WT and GS, Sed (left) and VWR (right) mice. (**A**,**E**) Analysis of significantly up (red) and downregulated (blue) proteins as compared to WT samples. (**B**,**F**) Volcano plot analysis of significantly altered proteins. (**C**,**G**) Top ten altered GO Cellular process pathway analysis from identified significantly altered proteins with (**D**,**H**) network analysis of top ten pathways. (**I**) Mitochondrion pathway heatmap of the 20 most significantly up- and down-regulated proteins within the pathway when comparing GS Sed vs. WT Sed (column 1) and GS VWR vs. WT VWR (column 2). The complete pathway of all significantly altered proteins is shown in Appendix A. For network analyses, connected nodes display 20% or more overlap of proteins between sets, thicker connections display increased overlap; darker nodes depict more significantly enriched protein sets; larger nodes depict larger protein sets. Only significantly altered proteins with *p* < 0.05 and a Log_2_ fold change > 1 were identified in the volcano plots. Only significantly altered proteins with *p* < 0.05 were included in pathway analysis. *n* = 3 for each cohort.

**Figure 6 cells-14-01383-f006:**
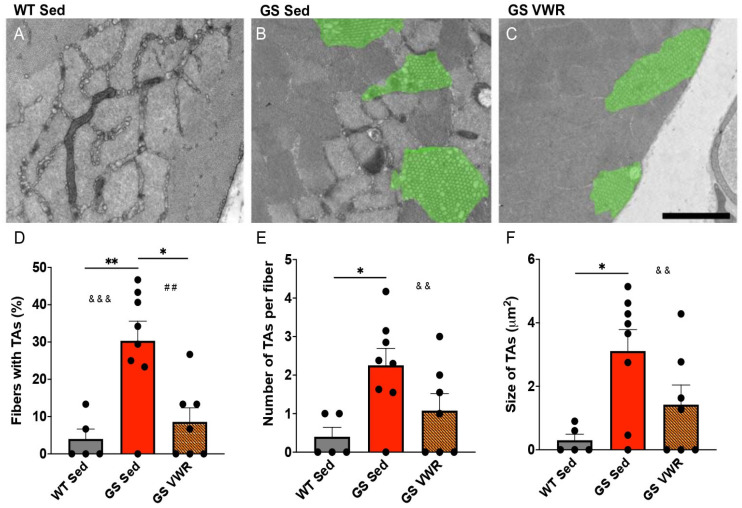
Assessment and quantification of EDL muscle TA prevalence and size. Transverse EDL muscle EM images from (**A**) WT Sed, (**B**) GS Sed, and (**C**) GS VWR mice. Green overlays denote TAs. Quantitative EM analysis of (**D**) percentage of fibers with TAs, (**E**) average number of TAs per fiber, and (**F**) average size of TAs. (**A**) Scale bar = 1.5 µm. (**B**,**C**) Scale bar = 2 µm. Each dot is representative of one mouse. Two-way ANOVA with multiple comparisons. * *p* < 0.05, **/##/&& *p* < 0.01, &&& *p* < 0.001. Isolated &, # denote ANOVA group effects of genotype and exercise, respectively. Data displayed as mean +/− S.E.M.

**Figure 7 cells-14-01383-f007:**
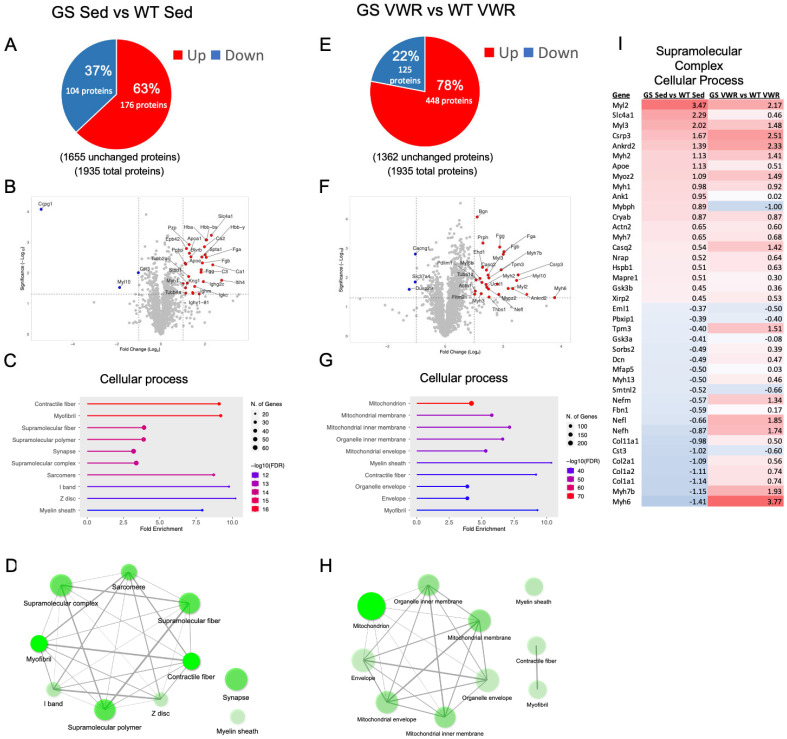
EDL proteomic analysis of WT and GS, Sed (left) and VWR (right) mice. (**A**,**E**) Analysis of significantly up (red) and downregulated (blue) proteins as compared to WT samples. (**B**,**F**) Volcano plot analysis of significantly altered proteins. (**C**,**G**) Top ten altered GO Cellular process pathway analysis from identified significantly altered proteins with (**D**,**H**) network analysis of top ten pathways. (**I**) Supramolecular complex pathway heatmap of the most significantly up- and down-regulated proteins within the pathway when comparing GS Sed vs. WT Sed (column 1) and GS VWR vs. WT VWR (column 2). The complete pathway of all significantly altered proteins is shown in Appendix A. For network analyses, connected nodes display 20% or more overlap of proteins between sets, thicker connections display increased overlap; darker nodes depict more significantly enriched protein sets; larger nodes depict larger protein sets. Only significantly altered proteins with *p* < 0.05 and a Log_2_ fold change > 1 were identified in the volcano plots. Only significantly altered proteins with *p* < 0.05 were included in pathway analysis. *n* = 3 for each cohort.

## Data Availability

All proteomic datasets will be deposited in the centralized PRIDE (ProteomicsIDEntifications) database. All other datasets in the current study are available from the corresponding author upon request.

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
