# Peer review of "Voluntary Wheel Running Mitigates Disease in an Orai1 Gain-of-Function Mouse Model of Tubular Aggregate Myopathy"

_cells, 2025, doi:10.3390/cells14171383_

Round 1

Reviewer 1 Report

Comments and Suggestions for Authors

The manuscript titled "Voluntary wheel running mitigates disease in an Orai1 gain-of-function mouse model of tubular aggregate myopathy." The abstract clearly summarizes the objectives, methods, and key findings of the study. The author presents the necessary details in several parts of the manuscript to an exceptional standard, although I have some comments regarding the following topics
Comments:

  • Figure 5 and Figure 7:
    Please improve the quality of these images to make them more accessible to the reader.
    In the current version, it is difficult to clearly see the contents.
    I recommend exporting the figures as PDFs and editing them in Illustrator, which may help enhance the image quality.
  • Mitochondrial-related proteins: Figure 5 & Figure 7
    I believe that proteins related to mitochondrial function may play an important role in this study.
    Could the authors clarify which specific proteins are involved?
  • Validation of findings:
    The statement about TA formation/mitigation (proteins involved in the formation of supramolecular complexes) and muscle dysfunction (proteins involved in controlling mitochondrial function) in TAM would be more convincing if validated using an additional methodology. This would strengthen the evidence and improve the overall impact of the study.
  • Did you observe that the level or intensity of voluntary wheel running varied between individual mice? How did you control or interpret those differences in the analysis?
  • How might your findings in this study (mouse model) translate to potential interventions in patients (human)?

Supplementary Information:

Since this study relies heavily on omics data, comprehensive supplementary materials are essential for a transparent and reproducible analysis. These should include:

  • Quality control metrics include data completeness (before and after normalization), reproducibility, and technical variability.
  • Full lists of proteins and differentially expressed proteins (DEPs), along with functional annotations.

Reviewer 2 Report

Comments and Suggestions for Authors

The work describes a well-conducted research on a still incompletely understood human muscle disease (TAM myopathy) modelled in Orai1 mice. The outcome of the work has been well documented in the original manuscript, supported by supplementary material, and largely discussed in a proper way with relevant literature references. The study design. methods and results originated from multidisciplinary laboratory analyses and the discussion is also relevant to the outcome of the animal study to better understand the human disease and to discuss a possible effective countermeasure (endurance exercise by running) over several weeks. There are no major comments from the reviewer´s point of view.

However, several minor comments and suggestions related to data presentation (mainly figure quality due to missing enlargements required) should be addressed by the authors:

  1. Fig.4: Please specify the black stars in B and C histology in legend.  Please also specify the black arrows in EM images shown in E and F also in the legend. Usually lipid droplets look like empty vacuoles marked here by the black arrows following preparation artifacts (ethanol extraction).
  2. In Fig 4, only ultra-sectional profiles of mitochondria are seen and no whole mitochondria (that could be counted in quantity). Quantification of ultrastrutures is hard to determine due to the high number of follow-up ultrasections in EM ultrasection preparations. So only numbers of mitochondrial profiles (and not numbers of mitochondra as organelles) can be calculated. This should be clarified and explained in the relevant text parts but also in the figure legends.
  3. Figs 5 and 7 are fine but show poor reading quality (text must be enlarged) 
  4. Fig. 6. Please specify the overlays in B and C in legend for the reader. Are these the accumulated TA profiles ?
  5. The possibility of sex-related differences should be discussed (study limitation) given the fact that TAM is more expressed in male than in female humans modelled by the GS mice used in this work. The authors should mention why both male and female mice were included in their test groups, perhaps also highlighted in a separate Discussion part (study limitations).

Round 2

Reviewer 1 Report

Comments and Suggestions for Authors

Ten days should be sufficient to perform targeted proteomics to support the reliability of the candidate proteins identified in this study. This is especially important given the risk of false positives among the differentially expressed proteins (DEPs); the current statistical threshold (p < 0.05 and fold change >1.5) increases this risk in the absence of further validation. Moreover, there is minimal change observed, particularly in the soleus proteomic analysis of WT and GS (Figure 5).

If completing this within ten days is not feasible, I believe Cells allows for extension requests to ensure higher-quality research.
